# Lipid Antigens: Revealing the Hidden Players in Adaptive Immune Responses

**DOI:** 10.3390/biom15010084

**Published:** 2025-01-08

**Authors:** Tamana Eskandari, Yasamin Eivazzadeh, Fatemeh Khaleghinia, Fatemeh Kashi, Valentyn Oksenych, Dariush Haghmorad

**Affiliations:** 1Student Research Committee, Semnan University of Medical Sciences, Semnan 35147-99442, Iran; 2Department of Immunology, School of Medicine, Semnan University of Medical Sciences, Semnan 35147-99442, Iran; 3Faculty of Medicine, University of Bergen, 5020 Bergen, Norway

**Keywords:** adaptive immune system, lipid antigen, natural killer, NKT cells, CD1, lipidomics

## Abstract

Traditionally, research on the adaptive immune system has focused on protein antigens, but emerging evidence has underscored the essential role of lipid antigens in immune modulation. Lipid antigens are presented by CD1 molecules and activate invariant natural killer T (iNKT) cells and group 1 CD1-restricted T cells, whereby they impact immune responses to pathogens and tumors. Recent advances in mass spectrometry, imaging techniques, and lipidomics have revolutionized the identification and characterization of lipid antigens and enhanced our understanding of their structural diversity and functional significance. These advancements have paved the way for lipid-based vaccines and immunotherapies through the application of nanoparticles and synthetic lipid antigens designed to boost immune responses against cancers and infectious diseases. Lipid trafficking, CD1 molecule interactions, and the immune system’s response to lipid antigens are yet to be completely understood, particularly in the context of autoimmunity and microbial infections. In the years to come, continued research efforts are needed to uncover its underlying biological mechanisms and to exploit the full potential of therapies directed against lipid antigens.

## 1. Introduction

The adaptive immune system, which integrates antigen-presenting cells (APCs), B cells, and T cells, produces specific immunity to antigens via cellular and humoral immune responses [1]. Contrary to the innate immune system, it also bestows immunological memory. After a primary response, antibodies produced by long-lived plasma cells in the bone marrow and memory B and T cells in secondary lymphoid tissues provide long-lasting immunity, allowing faster and more potent responses upon re-exposure [2].

Recognition of antigens presented by APCs through the T cell receptor (TCR) is essential for cellular immune responses. T cells can be divided into different functional types based on environmental cues and co-stimulatory or inhibitory signals [3]. Cytotoxic CD8^+^ T cells, CD4^+^ T helper cells, regulatory T cells, and memory T cells perform diversified functions such as killing infected cells through the release of perforins and granzymes, providing co-stimulatory signals through direct cell contact or cytokine release, regulating immune responses to prevent tissue damage, and providing faster and more potent responses during re-exposure to antigens (Figure 1) [4,5].

Naïve B cells identify antigens through their BCR and after co-stimulatory interactions with CD4^+^ T helper cells differentiate into plasma cells and memory B cells [6]. Plasma cells secrete antibodies that circulate throughout the body and tag foreign antigens for elimination.

## 2. Traditional Focus on Protein Antigens

A protein antigen may be recognized by the immune system and trigger an immune response (i.e., an immunogen). Protein antigens can originate from pathogens (such as viruses or bacteria), allergens derived from the environment, or from the body’s own tissues [7]. Protein antigens vary in structure: linear epitopes consist of short, continuous amino acid sequences whereas those forming conformational epitopes are recognized as three-dimensional structures. Protein antigens are essential in the development of vaccines, diagnostics, and therapies for a variety of diseases [8].

When the immune system encounters protein antigens, it triggers a series of responses aimed at neutralizing or eliminating the perceived threat. This includes the activation of T cells, the production of antibodies, and the generation of memory cells for future immunity [9]. The immune response to protein antigens is a complex and highly regulated process. Antigen processing and presentation are the cornerstones of adaptive immunity. Generally, B cells cannot generate high-affinity antibodies to protein antigens without cognate T cell help, which is typically provided in follicular germinal centers or through extrafollicular interactions [10,11]. CD4^+^ T cells, which provide such help, recognize short antigen-derived peptides bound to cell surface glycoproteins encoded by polymorphic major histocompatibility class II (MHC-II) genes presented by APCs (Figure 2) [12]. In contrast, eradication of virus-infected cells occurs through the action of cytotoxic CD8^+^ T cells, which rely on the recognition of peptide–MHC class I complexes for their action [13].

For research and diagnostic purposes, detection reagents with enhanced avidity for antigen-specific TCRs are essential. This is achieved by generating MHC class I–peptide tetramers through the polymerization of MHC molecules, which are then tagged (e.g., via biotinylation) to facilitate signal amplification and precise detection [14].

Researchers utilize a range of techniques to study protein antigens, including mass spectrometry, protein crystallography, ELISA, and various molecular biology methods. While traditional methods have been instrumental in protein antigen studies, they are not without limitations [15]. Technical challenges such as time-consuming protocols, limited detection sensitivity, and cross-reactivity have hindered a comprehensive understanding of certain antigens [16]. Recent years have witnessed significant advancements in protein antigen research, from high-throughput screening techniques to advanced computational modeling; these developments have revolutionized the study of antigens and enabled deeper insights into their structures, functions, and interactions with the immune system [17].

The protein-centric view of immune responses and molecular approaches based thereon have arguably been unfavorable to our understanding of carbohydrates, lipids, and nucleic acids as antigens [18,19]. The immune system is trained to recognize and respond to foreign antigens; nevertheless, it also needs to be tolerant of self-antigens to avoid autoimmune reactions [20]. Immune recognition also involves non-protein-sensing components such as pattern recognition receptors (PRRs) and certain Toll-like receptors (TLRs) which recognize conserved molecular patterns on pathogens. Such components are essential for efficient immune responses but are not directly addressed by protein antigen-based studies [21,22].

## 3. Introduction to Lipids in Adaptive Immunity

The immune system is stimulated by a wide range of antigenic molecules from both endogenous and external sources [23]. In addition to proteins, lipids, and phosphorylated metabolites can also activate T cells [24]. Lipids are hydrophobic biomolecules that play crucial roles in immune responses. They are essential components of cell membranes, influencing membrane properties and immune cell functions [25].

Lipids also serve as signaling molecules that regulate immune cell activation, differentiation, and effector functions. They can directly activate immune cells through their interactions with specific receptors or co-receptors, such as lipopolysaccharides (LPSs) on innate immune cells [26]. In B cells, signals through TLR4 and the BCR synergize [27]. To initiate adaptive immune responses, lipid antigens are presented by APCs to T helper cells via CD1 molecules [28].

Lipid antigens can originate from various sources, including bacteria, fungi, environmental factors (such as allergens), and endogenous lipid molecules present in host cells and tissues. Additionally, some viruses acquire lipids from host cells during their replication process, forming a lipid envelope [29]. The recognition of lipid antigens by the immune system involves microbial CD1d-restricted lipid-based antigens and NKT cells [30]. Atypical MHC class I proteins of the CD1 family bind and present lipid antigens. Compared to classical MHC class I molecules, which are highly polymorphic, CD1 genes are less diverse in terms of sequence, having very few or no synonymous single-nucleotide polymorphisms [31]. When the TCR of a lipid-specific T cell forms cognate contacts with CD1–antigen complexes on the APC, both cells become activated. This interaction is critical for invariant natural killer T (iNKT) cell maturation, selection, and egress from the thymus. It may also aid in the formation of group 1 CD1-restricted T cells. In the periphery, lipid antigens derived from the body, microbes, and the environment may activate CD1-restricted T lymphocytes (Figure 3) [32].

The quality and amount of complexes present on the surface of APC, TCR affinity, and the successful engagement of co-stimulatory and adhesion molecules with their corresponding ligands all have an impact on the stimulatory interaction of the TCR with CD1–antigen complexes [33]. The ability of CD1-restricted T lymphocytes to migrate through tissues is crucial for their activation. Depending on the strength of the signal received during antigen recognition (determined by the number of CD1 complexes), CD1-restricted T cells release a range of cytokines [34]. The inherent properties of lipids define the physical needs for their presentation mechanism. Lipids exhibit poor solubility in water, which means that they are consistently linked to membranes or lipid-transfer proteins (LTPs) in biological fluids and tissues [35]. Lipid-binding proteins are essential for lipid absorption by APCs, their intracellular transit, and processing prior to CD1 loading [36]. Furthermore, distinct CD1 isoforms exhibit both overlapping and disparate lipid-binding specificities, which directly affect the ability of lipid antigens to elicit T cell activation [30].

## 4. Discovery of Lipid Antigens

Unraveling the role of lipids as antigens has been a gradual process, marked by significant milestones that have deepened our understanding of immune recognition and immune responses. Two landmark papers described lipid presentation to T cells on CD1 molecules [37,38]. CD1 molecules are a group of glycoproteins structurally similar to MHC class I molecules. While MHC molecules are primarily involved in presenting peptide antigens to T cells, CD1 molecules possess a hydrophobic binding groove capable of accommodating lipid antigens [39]. This discovery provided the first insight into the existence of a specialized pathway for presenting lipid antigens to T cells [40]. Subsequent investigations demonstrated that lipid antigens, such as glycolipids and phospholipids, could indeed stimulate T cell responses. This finding challenged the conventional understanding that immune recognition was solely based on protein antigens [41]. Subsequent studies revealed the presence of specialized T cell subsets, including NKT cells and CD1-restricted T cells, which exhibited unique recognition properties and played critical roles in immune regulation and host defense [42].

The isolation and characterization of lipid antigens from various sources further expanded our understanding of their structural diversity and functional significance. Techniques such as lipid extraction, purification by HPLC, and mass spectrometry enabled researchers to identify specific lipid antigens and elucidate their roles in immune responses [43]. Functional studies using lipid antigens and genetically modified mouse models lacking CD1 molecules provided valuable insights into the mechanisms underlying lipid antigen presentation and immune modulation [28,36,44]. These studies demonstrated that lipid antigens could modulate immune cell activation, cytokine production, and effector functions, thereby influencing the outcome of immune reactions [45,46].

Moreover, the pathophysiological relevance of lipid antigens becomes apparent in various contexts, including infectious diseases, autoimmune disorders, and cancer [47,48,49]. Lipid antigens derived from microbial pathogens, such as mycobacteria and *Borrelia burgdorferi*, were found to activate immune responses and contribute to host defense mechanisms [50,51]. Conversely, dysregulation of lipid antigen recognition has been implicated in autoimmune diseases, highlighting the importance of lipid-mediated immune regulation in maintaining immune homeostasis [52].

## 5. Role of Lipid Antigens in T Cell Activation

Understanding lipid recognition by T cells is crucial for understanding host defenses to infections, tumor immune surveillance, and autoimmunity [30]. CD1 antigen-presenting molecules bind lipids to present them to the T cells. It follows that lipids are only T cell immunogenic if they can bind CD1 [53]. Five isoforms of CD1, non-classical MHC class I-like proteins with a narrow, deep hydrophobic cleft binding groove, are called CD1a-e in humans. CD1a-d present foreign and self-lipid/glycolipid antigens to lipid-reactive T lymphocytes, whereas CD1e only takes part in antigen processing but not in presentation [33,54]. T cell responses are only elicited after CD1 lipid presentation on the cell surface after the following steps: antigen uptake, cross-priming and cross-presentation, processing, CD1 assembly and recycling, lipid trafficking, CD1 loading with lipids, and persistence of lipid–CD1 complexes on the cell surface [55,56].

The uptake of lipid antigens primarily involves intracellular loading onto CD1 molecules, although surface loading may occur in specific cases. Glycolipids can activate specific T cells through cross-priming and cross-presentation, where antigens are internalized by CD1d-expressing APCs, processed, and presented. For instance, microbial glycosphingolipids undergo partial degradation in late endosomes, exposing their antigenic head groups before being loaded onto CD1 molecules for engagement with TCRs [57]. The assembly and recycling of CD1 molecules vary across isoforms, with CD1a predominantly trafficked through early recycling endosomes, while CD1b is processed in late endosomes, reflecting their distinct roles in antigen presentation within cellular compartments [58]. Lipid trafficking, mediated by lipid-transfer proteins, influences the distribution and immunogenicity of lipids, with acidic and neutral lipids continuously moving between membrane layers. In the following step, the persistence of lipid–CD1 complexes on APC surfaces is substantial for efficient priming of T cells. CD1b and CD1c have shorter half-lives due to transport in distinct endosomal compartments different from CD1a [59].

T cells recognizing lipids via CD1 molecules fall into two main groups: those restricted by group 1 CD1 molecules (i.e., CD1a-CD1c) and those restricted by CD1d, which resemble peptide-specific T cells and innate immune cells, respectively [60]. The primary CD1d-restricted cells are invariant natural killer T (iNKT) cells, which express NK cell markers and a semi-invariant TCR (an invariant TCRα chain paired with a TCRβ chain and limited Vβ usage) [33,54]. Upon detection of the lipid antigen, iNKT cells produce both Th1 (TNF-α and IFN-γ) and Th2 (IL-4, IL-5, and IL-13) cytokines; this dual functionality allows iNKT cells to support both inflammatory and immune-regulatory pathways. These cytokines rapidly induce DC maturation, which facilitates cytotoxic T cell priming [41,61]. It has also been demonstrated that iNKT cells can secrete anti-inflammatory cytokines such as TGF-β and IL-10 in response to polyclonal TCR activation.

Group 1 CD1-restricted CD4^+^ T cells, although diverse in their TCRs, respond more slowly to CD1–lipid complexes as compared to iNKT cells, which are more similar to conventional T cells. Most of these T cells recognize combined CD1–self-lipid complexes, while some recognize CD1 molecules alone [62]. Group 1 CD1-restricted T cells release a variety of cytokines in response to infection, inflammation, and tumors. When stimulated with self-antigens, they release IFN-γ, TNF-α, IL-17, and IL-22 [42]. The majority of them do not appear to release Th2 and/or regulatory cytokines in response to antigenic stimulation, in contrast to group 2 CD1-restricted T cells. More research is required to find out if Th2 and/or regulatory cytokines are produced by group 1 CD1-restricted T cells in different infectious or inflammatory settings [61,63,64].

## 6. Lipids in B Cell Responses

Lipids play essential roles in B cell responses, influencing various aspects of B cell activation, differentiation, and antibody production [65]. Lipids can directly influence B cell activation through interactions with specific receptors or co-receptors on the B cell surface. For example, certain lipid antigens or lipid-derived molecules can bind to TLRs on B cells, triggering signaling pathways that lead to B cell activation and proliferation [66]. Additionally, lipid rafts, specialized microdomains enriched in cholesterol and sphingolipids, play a crucial role in BCR signaling by facilitating the clustering of BCRs and downstream signaling molecules upon antigen recognition [67,68].

Lipids contribute to the process of antibody production by providing essential components for B cell function. Lipids are crucial for the formation and stability of lipid bilayers in membranes, including the membranes of plasma cells, which are responsible for antibody secretion [67]. Furthermore, lipids serve as precursors for lipid mediators that can modulate B cell function and antibody production. For instance, certain eicosanoids derived from arachidonic acid metabolism have been shown to regulate B cell proliferation and antibody secretion [69].

Specific CD1 molecules, such as CD1c and CD1d, are expressed on specific subsets of B cells, including marginal zone (MZ) B cells and memory-like B cells. These molecules play crucial and unique roles in lipid antigen presentation, distinguishing their function from MHC molecules, which primarily present peptides. CD1c and CD1d facilitate the presentation of lipid and glycolipid antigens to CD1-restricted T cells, enabling unique interactions in the immune system. While CD1d predominantly engages iNKT cells, key players in innate immunity, CD1c may interact with other specialized T cell subsets. However, its exact role requires further exploration [70,71].

CD1d expression enables B cells to present lipid antigens and activate NKT cells, leading to rapid cytokine secretion that enhances or modulates immune responses. In contrast, CD1c, highly expressed in MZ B cells, is implicated in broader immune regulation through its ability to present antigens to T cells. However, the expression of these molecules is not static; B cell activation, mainly through CD40L signaling, downregulates both CD1c and CD1d. This dynamic regulation, which creates a limited window for optimal lipid antigen presentation, is an intriguing and engaging aspect of B–T cell interactions in immune regulation that warrants further exploration [70,71].

Notably, retinoic acid receptor (RAR) signaling pathways can reverse the downregulation of CD1d and CD1c induced by activation, suggesting a potential therapeutic target for modulating immune responses. These findings underscore the paramount importance of CD1 molecules in shaping humoral immunity and reveal a complex regulatory mechanism that balances B cell activation with lipid antigen presentation to maintain immune homeostasis. The potential therapeutic implications of these findings in modulating immune responses highlight the practical applications of this research and significantly impact our understanding of immune regulation [71,72].

Lipids may also play a role in antibody class switching, a process by which B cells change the class of antibody they produce, e.g., switching from IgM to IgG [73]. Lipid mediators, such as prostaglandins, have been shown to regulate antibody class switching by modulating cytokine production and B cell activation [74]. Additionally, lipid antigens presented by APCs may influence the cytokine microenvironment, thereby affecting the direction of antibody class switching. Understanding the intricate interplay between lipids and B cells is essential for elucidating the mechanisms underlying immune responses and may offer insights into the development of novel therapeutic strategies targeting lipid pathways in immune-related diseases [75].

## 7. Impact of Lipid Antigens on the Immune System and Its Relevance to Disease

The identification of lipid molecules by T lymphocytes as antigens was a groundbreaking finding in the field of cellular immunology, leading to novel insights into the immune response against microorganisms, cancer cells, and autoimmunity [37]. Immune-mediated illnesses can develop or worsen as a result of any disturbance to the balance that governs lipid homeostasis in adaptive immune cells [76].

Lipid antigens, including glycolipids and phospholipids, can influence cancer progression by modulating immune responses and affecting the tumor microenvironment. Dysregulation of lipid metabolism is a hallmark of cancer [77]. Highly expressed glycosphingolipids (GSLs) are tumor-associated antigens that trigger antibody responses [78]. GSLs play a role as adhesion molecules for tumor cells during metastasis and are involved in signal transduction. GSLs are exploited to develop antitumor vaccines and are potential targets for cancer therapy [79]. Normal tissue and tumors both contain GSLs, but only when these lipids are on the surface of tumor cells can they function as effective immunogens. In contrast to GSLs found on normal cells, tumor-derived GSLs might react differently to antibodies [80]. GSLs are thought to regulate important molecules involved in signal transduction, specifically gangliosides and the breakdown products of these molecules, to regulate cell development [81]. Patients with melanoma have ganglioside-specific serum antibodies. Melanoma growth may be inhibited by actively immunizing against gangliosides to artificially raise the level of specific anti-ganglioside antibodies [82].

The presentation of *M. tuberculosis* lipid antigens by group 1 CD1 molecules has been extensively studied. In vitro investigations have revealed that group 1 CD1-restricted T cells exhibit cytotoxicity and produce IFN-γ and TNF-α upon encountering *M. tuberculosis* antigens [83]. Furthermore, individuals exposed to mycobacteria show higher frequencies of group 1 CD1-restricted *M. tuberculosis* lipid antigen-specific T cells compared to a control population [84,85]. *M. tuberculosis* produces a diverse array of lipids, which form stable complexes with CD1 molecules and elicit T cell responses. The structures of these antigenic lipids can vary greatly; each lipid may stimulate unique T cells that are capable of discerning subtle lipid structural changes [86,87]. Certain *M. tuberculosis* lipid antigens play crucial roles in preventing the emergence of negative mutants capable of evading the immune response. T cells specific for lipid antigens are activated during *Mycobacterium tuberculosis* infection and contribute to protective immune functions [88].

Several studies have shown that the LPS of some *C. jejuni* strains causes people with Guillain–Barre syndrome (GBS) to produce antibodies that cross-react with gangliosides from peripheral nerves [89].

Sphingolipids in host cells are used as membrane receptors by a variety of pathogens. Pathogen infection and host defense are regulated by sphingolipid metabolites. For example, a particular glycosphingolipid functions as an endogenous ligand to activate NKT cells [90]. Furthermore, the development of antibodies that cross-react with mammalian sphingolipids is a contributing factor in certain autoimmune disorders that arise after infections [48].

## 8. Lipid-Based Vaccines and Immunotherapies

Lipid-based immunotherapies, also known as developmental vaccines and cancer immunotherapies, regulate immune system responses by using lipid molecules in the form of adjuvants or vectors. Several lipid-based techniques have been explored in recent years to modulate immune responses and aid in disease treatment [91].

One of these strategies is the use of lipid-based nanoparticles (LBNPs) consisting of liposomes, emulsions, solid lipid nanoparticles, and nanostructured lipid carriers, which can carry and encapsulate immunomodulatory compounds such as small molecules (synthetic drugs) and nucleic acids (e.g., mRNA [92]). Compared to conventional delivery strategies, this new strategy has higher solubility in the aqueous phase and lower toxicity. It also aids the delivered nucleic acid to be released from endosomes and increases the stability of its content [93]. Nevertheless, it has disadvantages such as liver toxicity, cationic lipid cytotoxicity, and a low efficacy of small molecule encapsulation. Thus, we need to put further effort into improving LBNP technology. To increase vaccine efficacy against pathogens, adjuvant strategies employ lipid-based delivery systems such as liposomes and emulsions [94].

Apart from the previously mentioned techniques, synthetic lipid antigens are also employed to modulate immune responses. For example, recent studies have demonstrated that synthetic glycolipids, either by themselves or loaded on DCs, can trigger the production of pro-inflammatory cytokines by iNKT cells. This results in an antitumor effect, through the stimulation of cytotoxic lymphocytes with specificity for tumor antigens and through the inhibition of angiogenesis [95,96]. Similarly, tumor growth inhibition can be achieved through activating NKT cells by the use of the CD1d/glycolipid combination and cancer-specific antibody complex. These applications promise novel techniques in immunotherapies by using lipid antigens to modulate immune responses, leading to fighting a variety of diseases such as cancers and autoimmune disorders [97].

## 9. Technological Advances in Lipid Antigen Research

Lipids play critical roles in biological systems; thus, deepening our understanding of lipid metabolism may provide novel insights into the diagnosis and pathophysiology of diseases. Recent technological advancements have enabled the development of cutting-edge tools for lipid antigen analysis. Mass spectrometry-based techniques, such as MALDI-TOF and ESI-MS, have revolutionized the identification and characterization of lipid antigens [98]. Additionally, high-resolution imaging modalities, including super-resolution microscopy and cryo-electron microscopy, have provided unprecedented insights into lipid antigen interactions within cellular contexts. These emerging tools offer enhanced sensitivity, resolution, and reproducibility, empowering researchers to unravel the complexities of lipid antigen structures and functions with unprecedented precision [99].

Recent advances in lipidomics, mainly through advanced mass spectrometry techniques, have significantly enhanced our understanding of lipid antigen presentation by CD1 molecules [43]. High-performance liquid chromatography coupled with quadrupole time-of-flight mass spectrometry (HPLC-QToF-MS) has enabled sensitive detection and characterization of a wide range of lipids bound to CD1 molecules, including hydrophobic lipids, glycolipids, and phospholipids [100]. This approach allows for the resolution of ion chromatograms representing diverse lipid species, while collision-induced dissociation (CID)–MS further elucidates their chemical structures. These methods have facilitated the identification of specific lipid-binding motifs and size-dependent antigen display mechanisms across different CD1 isoforms. For instance, structural studies revealed that CD1b can bind two small lipids simultaneously within its cleft, unlike the single-lipid binding observed for CD1a and CD1d, highlighting isoform-specific lipid presentation strategies [101].

Additionally, the integration of lipidomics and bioinformatics has led to the comprehensive mapping of over 2000 lipid species presented by CD1 molecules. This extensive lipidomic resource, offering a wealth of information for identifying new antigens and therapeutic targets, has the potential to revolutionize biomedical research. High-resolution crystallography has further contributed by providing detailed insights into lipid seating mechanisms, such as the dual-chamber arrangement in CD1b that positions lipids for effective T cell receptor interaction [101]. These advances deepen our understanding of the molecular basis of lipid antigen presentation and have implications for biomedical research, including the identification of lipid-based immune regulators and potential interventions for diseases involving lipid-mediated immunopathology [102].

Lipidomics, driven by advancements in mass spectrometry and chromatography, offers deep insights into disease diagnosis and pathophysiology as well. Integrating lipidomics with other biological data sources holds promise for drug development and will indisputably provide clinicians with novel insights into disease biology [103]. Artificial intelligence and machine learning have emerged as indispensable tools in lipid antigen discovery. Advanced algorithms can analyze vast datasets to identify potential lipid antigens, as well as to predict their binding affinities and assess their immunological relevance. The integration of AI technologies with experimental approaches has enabled lipid antigen discovery, thereby offering new avenues for the development of targeted therapeutics and precision medicine strategies [104].

## 10. Future Directions and Challenges

Understanding the limitations and opportunities of lipid antigen discovery is essential for advancing research and addressing critical issues regarding the role of lipids in immune responses. Current research on CD1-restricted T cell responses, which have the potential to cause skin, respiratory, and intestinal diseases, is limited, partly due to the challenges posed by working with infectious organisms and the absence of CD1 group 1 molecules in mice [104,105]. These responses, associated with various disorders, are further hindered by technological limitations in isolating and detecting lipid antigens. Nevertheless, advancements in CD1 tetramers, 3D-culture methods, and lipidomics offer promising avenues to elucidate CD1-mediated immune responses in the years to come. Furthermore, the high frequency of lipid-specific T cells, which are crucial to immune response modulation, is also associated with autoimmune diseases and defense against infections. Nevertheless, human CD1 polymorphisms preclude lipid-specific immunity, making microbial lipids less susceptible to genetic mutation-driven selection pressures [41,106].

In addition to clarifying the molecular mechanisms of lipid insertion into the CD1 groove, future research in lipid antigen biology should concentrate on defining the cellular sub-compartments where lipid antigens and CD1 proteins interact. Although CD1 protein trafficking has been the focus of previous research, it is imperative to examine lipid trafficking as well. The non-random distribution of lipids presented by CD1 in particular cellular sub-compartments implies complex mechanisms that should be investigated to gain a better understanding of the roles of lipid-specific T cells and their immunological therapeutic implications [107,108].

Recent advancements have shed light on the role of lipid antigens in T cell immunity, particularly through group 1 CD1 molecules. These efforts have been facilitated by CD1 tetramer technology, solving crystal structures of lipid-reactive TCRs, and studies of defined molecular variations enabling diverse lipid-based antigen accommodation [109]. However, further research is needed to explore the impact of these variations on the antigen repertoire, lipid presentation, and T cell immunity. The complexity of CD1-restricted T cell functions, especially when it comes to recognizing lipid antigens, underscores the necessity for continued investigation into lipid processing and presentation, to understand lipid-specific immune responses and to develop therapeutic interventions [110]. Exploring CD1-mediated immune responses in the pathophysiology of mucosal tissues and their connections to cellular metabolism may pave the way for innovative therapeutics targeting a range of disorders [54].

Ultimately, the development of vaccines relies on identifying lipid compounds that contribute to bacterial pathogenicity and leveraging the absence of functional CD1 polymorphisms. A prominent example could be a study of how the use of lipid antigens could enhance anti-mycobacterial immunity [111]. The research asserts the significance of the CD1e polymorphism in regulating microbial antigen processing and enhancing immune responses specific to lipids. Variations in CD1e residues may impact the antigen binding pocket, potentially influencing vaccine development [112]. Preliminary trials using a crude lipid extract from *M. tuberculosis* as a vaccine have shown promising results in reducing pathology and bacterial burdens in tuberculosis models [113]. Further research into immunization with individual synthetic lipids offers potential as innovative subunit vaccine components, highlighting the promising role of lipids in vaccine development.

## 11. Conclusions

In summary, the exploration of lipid antigens and their role in adaptive immunity has unveiled significant insights that challenge the protein-centric view of immune responses. Key findings highlight that lipid antigens, presented by CD1 molecules, play a crucial role in immune surveillance and response, particularly through the activation of iNKT cells and group 1 CD1-restricted T cells. This expands our understanding of antigen diversity and the mechanisms through which the immune system recognizes and responds to pathogens and tumors.

The importance of lipid antigens in adaptive immunity underscores the need to consider these molecules in both fundamental immunological research and clinical applications. Technological innovations in mass spectrometry, imaging techniques, and lipidomics have revolutionized the identification and characterization of lipid antigens, offering deeper insights into their structural and functional diversity. These advancements have enabled the development of lipid-based vaccines and immunotherapies, promising novel strategies for enhancing immune responses against cancers and infectious diseases. Therefore, boosting immune responses against lipids may synergize with “classical” immunotherapies targeted against mutated self- or foreign protein antigens.

## Figures and Tables

**Figure 1 biomolecules-15-00084-f001:**
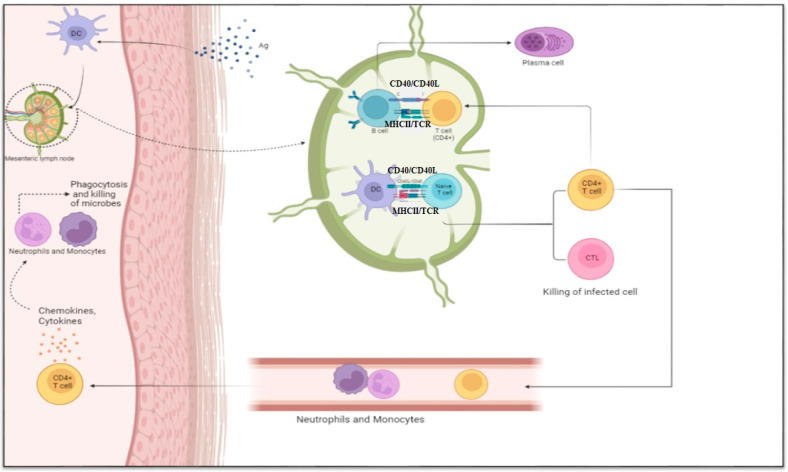
Adaptive immune system responses occur in the mesenteric lymph node. The antigen is presented to T lymphocytes by APCs (such as DCs or B cells), whereby they become activated and differentiate into several T cell types (CTLs and T helper cells). B cells differentiate into plasma cells and memory B cells that produce faster and stronger responses. The T cell’s production of chemokines and cytokines and its effects on neutrophils and macrophages are not described. The figure was designed using Biorender.com.

**Figure 2 biomolecules-15-00084-f002:**
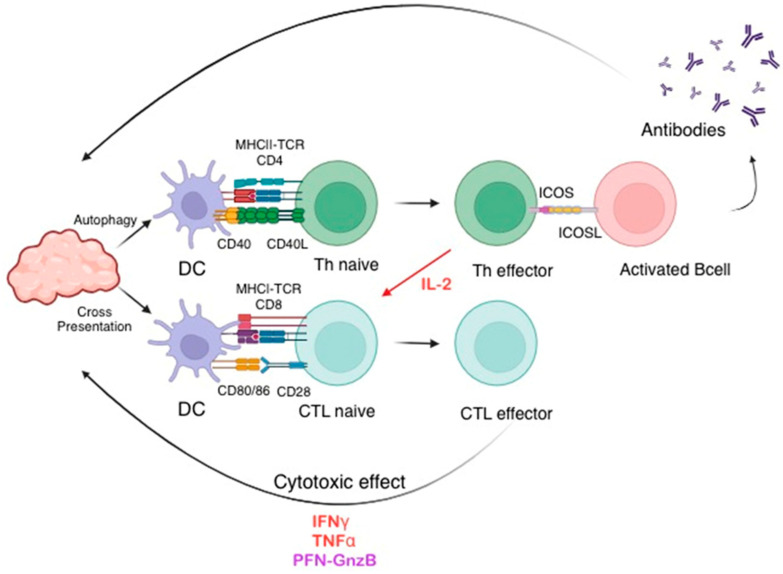
Tumor antigens activate CTLs through autophagy/MHCI for cytotoxicity and T helper cells through cross-presentation/MHCII for antibody production, contributing to tumor eradication through immune responses involving both cytotoxic and antibody-mediated mechanisms. The figure was designed using Biorender.com.

**Figure 3 biomolecules-15-00084-f003:**
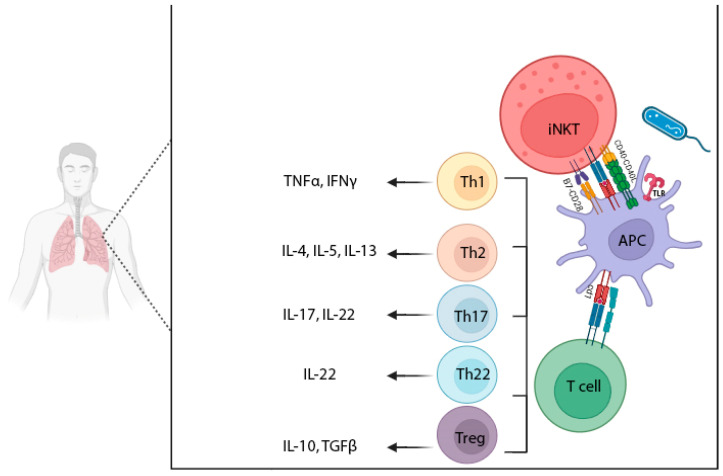
After the body’s encounter with the pathogen and recognition of bacterial lipid antigens by TLRs on the APC, lipid antigens are picked up and processed by APCs and presented on CD1 molecules to T cells and iNKT. Naïve CD4^+^ T cells differentiate into Th1 cells (which secrete TNF-α and IFN-γ), Th2 cells (which secrete IL-4, IL-5, and IL-13), Th17 cells (which secrete IL-17 and IL-22), Th22 cells (which secrete IL-22), and Treg cells (which secrete IL-10 and TGF-β). The figure was designed using Biorender.com.

## Data Availability

Not applicable.

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
