# Peer review of "Lipid Antigens: Revealing the Hidden Players in Adaptive Immune Responses"

_biomolecules, 2025, doi:10.3390/biom15010084_

Round 1

Reviewer 1 Report

Comments and Suggestions for Authors

Eskandari, et al have prepared an excellent review covering major aspects of research from the last 20 years into lipid antigens and their role in immunity. The review article adds context to lipid antigens in innate and adaptive immunity, as well as presentation by the CD1 family of antigen presenting molecules, and their downstream roles in adaptive immune responses in disease. Further, Eskandari et al cover the role of lipids as therapeutic design, the recent technological advances in lipid research, and raise critical future directions and challenges that have been present in the field for some time. The review article is well structured and well written, and extensively referenced, and will be of great benefit to the field. However, there are two sections that could benefit with some additional content, as some details may have been overlooked, or were not detailed enough: 
Section 6. Lipids in B cell response: It would be worth noting that some CD1 molecules (CD1c and CD1d) can be expressed on specific subsets of B cells, with roles in lipid presentation. Some references for your consideration: https://doi.org/10.4049/jimmunol.1003615, https://doi.org/10.3390/ijms23063372
Section 9: Technological advances in lipid antigen research: Specifically on lipidomics, there are recent studies that use lipidomics mass spectrometry to characterise lipid antigens directly eluted from CD1 molecules. As this section doesn't go into specific details on lipid antigens and CD1 molecules, and makes only sweeping statements, it would be important to include specific details and examples on recent technological advances to maintain context of this section with the rest of the manuscript. References to include: https://doi.org/10.1016/j.cell.2023.08.022

Minor corrections:
Line 46/139: Text in figures 1 and 3 are small and hard to read. Recommend increase image resolution in final manuscript, or if this does not improve the text quality, increase text size. 
Line 140: Remove word "on", so that figure legend reads "...bacterial lipid antigens by TLRs on the APC..."
Line 177: Either title of section "5. Introduction" is incorrect/incomplete, or this section is in the wrong place in the manuscript. 

Author Response

We thank the reviewer for the evaluation and constructive feedback.

Eskandari, et al have prepared an excellent review covering major aspects of research from the last 20 years into lipid antigens and their role in immunity. The review article adds context to lipid antigens in innate and adaptive immunity, as well as presentation by the CD1 family of antigen presenting molecules, and their downstream roles in adaptive immune responses in disease. Further, Eskandari et al cover the role of lipids as therapeutic design, the recent technological advances in lipid research, and raise critical future directions and challenges that have been present in the field for some time. The review article is well structured and well written, and extensively referenced, and will be of great benefit to the field. However, there are two sections that could benefit with some additional content, as some details may have been overlooked, or were not detailed enough: 

Question 1. Section 6. Lipids in B cell response: It would be worth noting that some CD1 molecules (CD1c and CD1d) can be expressed on specific subsets of B cells, with roles in lipid presentation. Some references for your consideration: https://doi.org/10.4049/jimmunol.1003615, https://doi.org/10.3390/ijms23063372

Response 1. We have incorporated this point into the revised manuscript, along with the provided references, to highlight the contribution of B cell subsets in lipid antigen presentation.

Question 2. Section 9: Technological advances in lipid antigen research: Specifically on lipidomics, there are recent studies that use lipidomics mass spectrometry to characterize lipid antigens directly eluted from CD1 molecules. As this section doesn't go into specific details on lipid antigens and CD1 molecules, and makes only sweeping statements, it would be important to include specific details and examples on recent technological advances to maintain context of this section with the rest of the manuscript. References to include: https://doi.org/10.1016/j.cell.2023.08.022

Response 2. This important point will provide greater depth and context to this Section. We have revised this section to incorporate these details and include the suggested reference to ensure alignment with the manuscript's overall focus.

Minor corrections:
Q3. Line 46/139: Text in figures 1 and 3 are small and hard to read. Recommend increase image resolution in final manuscript, or if this does not improve the text quality, increase text size. 

R3. We increased the text size within the figures to ensure readability.

Q4. Line 140: Remove word "on", so that figure legend reads "...bacterial lipid antigens by TLRs on the APC..."

R4. We removed the word "on" in figure legend

Q5. Line 177: Either title of section "5. Introduction" is incorrect/incomplete, or this section is in the wrong place in the manuscript. 

R5. We corrected the section title.

Reviewer 2 Report

Comments and Suggestions for Authors

The article by Eskandari et al describes the antigens recognised by T cells with a particular focus on lipid antigen presentation. They briefly summarise T cell recognition of peptide antigens and they then describe the importance of lipids in cellular responses before discussing CD1 antigen presentation and the T cell responses that are restricted by CD1.

This is a decent review article but it requires attention throughout and here are the key sections that require close attention:

35 TCR is not introduced i.e. T cell receptor

87-80 Need to re-write the tetramer section, there should not be a recombinant TCR involved, and this should be MHC class 1 not 2.

113 do viruses have lipids? Please check and confirm as this statement does not sound correct, with some viruses acquiring lipids from host cells

Fig 3 iNKT give rise to Treg or Th2. I suppose you mean that iNKT can be categorised into regulatory iNKT? And Th2 iNKT? However, iNKT generally have a mixed phenotype i.e. Th1 and Th2 and they produce a lot of IFNg and TNFa which is something that is mentioned later on in the manuscript. This figure is misleading and should be corrected.

177 title is introduction. Is this correct? This is confusing since there is an introduction in the beginning of the article. This should be changed.

191 not clear what this means, cross-priming and cross-presentation

192 lipid degradation statement. They should give specific detail on this with an example, because I don’t think this is common

207 should be TNFα

241-245 This sections is really not clear, how are the B cells activated, needs to be explained better

281 ref 85 is a thesis

Can use the following refs:

Moody, D. B., et al. (2000). "CD1c-mediated T-cell recognition of isoprenoid glycolipids in Mycobacterium tuberculosis infection." Nature 404(6780): 884-888.

Montamat-Sicotte, D. J., et al. (2011). "A mycolic acid-specific CD1-restricted T cell population contributes to acute and memory immune responses in human tuberculosis infection." J Clin Invest 121(6): 2493-2503.

284 should have refs:

Chancellor, A., et al. (2017). "CD1b-restricted GEM T cell responses are modulated by Mycobacterium tuberculosis mycolic acid meromycolate chains." Proc Natl Acad Sci U S A 114(51): E10956-E10964.

And

Van Rhijn, I., et al. (2017). "CD1b-mycolic acid tetramers demonstrate T-cell fine specificity for mycobacterial lipid tails." Eur J Immunol 47(9): 1525-1534.

372 ref 105 is a thesis not a published article

Other comments

T cells are written as T-cell or T cell, please choose one and be consistent

Figure 1, the TCR MHC II complex should be swapped i.e., TCR MHCII complex to reflect the position of the MHC on the APC and the TCR on the T cell above. Also CD40L should be on the T cell and visa versa

Author Response

We thank the reviewer for the evaluation and constructive feedback.

The article by Eskandari et al describes the antigens recognised by T cells with a particular focus on lipid antigen presentation. They briefly summarise T cell recognition of peptide antigens and they then describe the importance of lipids in cellular responses before discussing CD1 antigen presentation and the T cell responses that are restricted by CD1.

This is a decent review article but it requires attention throughout and here are the key sections that require close attention:

Q1. 35 TCR is not introduced i.e. T cell receptor

A1. The term "TCR" (T cell receptor) is introduced and defined upon its first mention in the manuscript.

Q2. 78-80 Need to re-write the tetramer section, there should not be a recombinant TCR involved, and this should be MHC class 1 not 2.

A2. We have now revised the tetramer section and highlighted it.

Q3. 113 do viruses have lipids? Please check and confirm as this statement does not sound correct, with some viruses acquiring lipids from host cells

A3. It is correct that viruses do not inherently possess lipids but acquire lipid envelopes from host cells during replication or budding. We revised the statement to clarify that the lipids associated with certain viruses are derived from the host cell membrane.

Q4. Fig 3 iNKT give rise to Treg or Th2. I suppose you mean that iNKT can be categorised into regulatory iNKT? And Th2 iNKT? However, iNKT generally have a mixed phenotype i.e. Th1 and Th2 and they produce a lot of IFNg and TNFa which is something that is mentioned later on in the manuscript. This figure is misleading and should be corrected.

A4. We have now revised the figure and legend to improve clarity.

Q5. 177 title is introduction. Is this correct? This is confusing since there is an introduction in the beginning of the article. This should be changed.

A5. We have now corrected the section title.

Q6. 191 not clear what this means, cross-priming and cross-presentation

A6. Specifically, cross-priming refers to the process by which APCs activate naïve CD8+ T cells by presenting extracellular antigens via MHC class I molecules. Cross-presentation is the mechanism underlying exogenous antigens are processed and presented on MHC class I molecules. We have now revised the text accordingly and highlighted it.

Q7. 192 lipid degradation statement. They should give specific detail on this with an example, because I don’t think this is common

A7. We have now revised the statement to provide specific details on lipid degradation in the context of antigen processing.

Q8. 207 should be TNFα

A8. It is now corrected and highlighted.

Q9. 241-245 This sections is really not clear, how are the B cells activated, needs to be explained better

A9. This paragraph has been removed during the revision.

Q10. 281 ref 85 is a thesis

Can use the following refs:

Moody, D. B., et al. (2000). "CD1c-mediated T-cell recognition of isoprenoid glycolipids in Mycobacterium tuberculosis infection." Nature 404(6780): 884-888.

Montamat-Sicotte, D. J., et al. (2011). "A mycolic acid-specific CD1-restricted T cell population contributes to acute and memory immune responses in human tuberculosis infection." J Clin Invest 121(6): 2493-2503.

A10. The mentioned reference was replaced.

Q11. 284 should have refs:

Chancellor, A., et al. (2017). "CD1b-restricted GEM T cell responses are modulated by Mycobacterium tuberculosis mycolic acid meromycolate chains." Proc Natl Acad Sci U S A 114(51): E10956-E10964.

And

Van Rhijn, I., et al. (2017). "CD1b-mycolic acid tetramers demonstrate T-cell fine specificity for mycobacterial lipid tails." Eur J Immunol 47(9): 1525-1534.

A11. The mentioned references have now been included.

Q12. 372 ref 105 is a thesis not a published article

A12. The mentioned reference was removed.

Other comments

Q13. T cells are written as T-cell or T cell, please choose one and be consistent.

A13. We have now changed "T-cell" to "T cell".

Q14. Figure 1, the TCR MHC II complex should be swapped i.e., TCR MHCII complex to reflect the position of the MHC on the APC and the TCR on the T cell above. Also CD40L should be on the T cell and visa versa

A14. It has now been corrected.

Round 2

Reviewer 2 Report

Comments and Suggestions for Authors

I have no further comments